# Enhancing Amyloid PET Quantification: MRI-Guided Super-Resolution Using Latent Diffusion Models

**DOI:** 10.3390/life14121580

**Published:** 2024-12-01

**Authors:** Jay Shah, Yiming Che, Javad Sohankar, Ji Luo, Baoxin Li, Yi Su, Teresa Wu

**Affiliations:** 1School of Computing and Augmented Intelligence, Arizona State University, Tempe, AZ 85281, USA; jgshah1@asu.edu (J.S.); yche14@asu.edu (Y.C.); baoxin.li@asu.edu (B.L.); teresa.wu@asu.edu (T.W.); 2ASU-Mayo Center for Innovative Imaging, Arizona State University, Tempe, AZ 85287, USA; 3Banner Alzheimer’s Institute, Banner Health, Phoenix, AZ 85006, USA; j.sohankar@bannerhealth.com (J.S.); ji.luo@bannerhealth.com (J.L.)

**Keywords:** partial volume correction (PVC), positron emission tomography, amyloid, deep learning, diffusion models, medical image super-resolution

## Abstract

Amyloid PET imaging plays a crucial role in the diagnosis and research of Alzheimer’s disease (AD), allowing non-invasive detection of amyloid-β plaques in the brain. However, the low spatial resolution of PET scans limits the accurate quantification of amyloid deposition due to partial volume effects (PVE). In this study, we propose a novel approach to addressing PVE using a latent diffusion model for resolution recovery (LDM-RR) of PET imaging. We leverage a synthetic data generation pipeline to create high-resolution PET digital phantoms for model training. The proposed LDM-RR model incorporates a weighted combination of L_1_, L_2_, and MS-SSIM losses at both noise and image scales to enhance MRI-guided reconstruction. We evaluated the model’s performance in improving statistical power for detecting longitudinal changes and enhancing agreement between amyloid PET measurements from different tracers. The results demonstrate that the LDM-RR approach significantly improves PET quantification accuracy, reduces inter-tracer variability, and enhances the detection of subtle changes in amyloid deposition over time. We show that deep learning has the potential to improve PET quantification in AD, effectively contributing to the early detection and monitoring of disease progression.

## 1. Introduction

Amyloid imaging is a crucial tool in the diagnosis and research of Alzheimer’s disease (AD). It allows for the non-invasive detection of amyloid-β (Aβ) plaques in the brain, which is a core neuropathological feature of AD [1]. Detecting Aβ pathology at the earliest stages of AD, before the onset of clinical symptoms, is critical for understanding disease progression, developing intervention techniques, and potentially improving patient outcomes. However, accurate quantification of amyloid using positron emission tomography (PET) imaging is limited due to the low spatial resolution of PET scans [2], which is typically around 5 mm and varies across scanner models and reconstruction algorithms. This causes the partial volume effect (PVE) [3], resulting in a spill-out of signal from target gray matter regions and spill-in from surrounding areas. The magnitude of the PVE depends on the size of the target region relative to the spatial resolution of the scans. In the context of amyloid PET imaging, the size of the target regions varies across subjects and often decreases as the subject ages or with disease progression. Therefore, PVE reduces the accuracy, precision, and statistical power of quantitative amyloid PET measurements. Another well-recognized issue of amyloid PET imaging is harmonizing data acquired using different scanners, tracers, and analytical pipelines. To minimize inter-scanner variabilities, a scanner-specific harmonization filter is often applied at the cost of further reduced spatial resolution [4]. To minimize the variability of amyloid PET measurements from different analytical pipelines, acquisition protocols, and tracers, the Centiloid scale was defined to linearly transform a particular measurement to this scale [5]. However, this Centiloid approach is designed for standardizing global measures and does not improve the between-measure agreements in terms of their shared variance [6,7,8]. We hypothesize that effective methods for spatial resolution recovery will improve PET quantification and reduce inter-tracer variabilities in amyloid PET measurements, and in this research, we propose a deep learning approach to achieve the goal.

Several partial volume correction (PVC) methods have been proposed in the literature to mitigate the PVE issue using anatomical information from MRI and CT [2,9,10,11,12,13,14,15]. Different from that, Tohka and Reilhac [16] showed that Richardson–Lucy, an iterative deconvolution-based method to recover spatial resolution in PET imaging and an alternative to MR-based approaches, offered comparable accuracy with reduced sensitivity to registration and segmentation errors. However, deconvolution-based correction methods are shown to amplify the image noise [17]. Different correction methods can also produce varying results, making standardization and comparison across studies challenging. Deep learning-based techniques [18,19] have recently been explored to tackle some of these challenges. Deep models can better learn complex patterns of tissue heterogeneity and can perform image denoising, potentially addressing noise amplification issues [19]. Deep models trained on diverse datasets may generalize better to different scanners and acquisition protocols [18], potentially improving the standardization and consensus among multi-center studies.

Instead of focusing on partial volume correction directly, as the PVC methods reviewed above do, an alternative is image super-resolution (SR), which refers to the task of rendering a high-resolution image from its low-resolution counterpart. We contend that PVE may be tackled during the process of rendering high-resolution PET from low-resolution PET. SR is a well-studied research problem in computer vision and image processing [20,21]. Use cases of SR span a broad spectrum, improving existing computer vision tasks [22,23,24] by improving image spatial resolution and perceptual quality, improving surveillance [25], and enhancing diagnostic accuracy in medical research using different imaging modalities [26,27,28]. Traditional methods for image SR heavily rely on image statistics [29,30,31], which has been shown to generate blurry and noisy artifacts in their high-resolution outputs [20]. With the advent of deep learning algorithms, several end-to-end architectures have been proposed where the models learn the mapping of low-resolution to high-resolution images through regression-based learning [32]. However, these methods fail to recover high-frequency details mainly because they learn an average mapping from the training dataset (due to L_1_/L_2_ loss functions), resulting in overly smooth model outputs and lacking spatial details [21].

To address these limitations, generative models have been explored for SR in recent years. Generative models learn to transform a latent variable *z* with a tractable prior distribution to a learned data space. Generative adversarial networks (GANs), flow-based methods, and diffusion models are three common generative models used to generate synthetic data. They differ in their core approach: GANs [33] are trained in an adversarial setting with generator and discriminator networks, flow-based methods [34] rely on invertible transformations to manipulate data distributions, while diffusion models [35] iteratively add and then learn to remove noise to generate data. GANs are known to suffer from mode collapse [36], resulting in unstable training and limiting the diversity of generated samples. Flow-based methods can impose topological constraints on the mapping between latent and data spaces, limiting their flexibility in modeling complex data distributions [37]. Except for longer sampling times, diffusion models have shown superior performance in generating high-fidelity medical imaging datasets [38,39,40]. Unlike GAN priors, diffusion model priors can preserve more information to generate realistic data. Motivated by this, we propose an SR solution based on the diffusion model to improve PET quantification.

The success of any SR methods (e.g., GAN, diffusion) heavily relies on the quantity and quality of the training data. Synthetic data has been substantially useful in medical AI research to alleviate issues such as a lack of datasets, annotations, privacy concerns, and high acquisition costs [41,42,43]. Data samples are typically artificially generated using domain knowledge or modeling techniques to mimic the characteristics and structure of real data without being directly derived from actual observations. It can be used to train AI models where target data are unavailable or scarce and provides a promising alternative to making AI models generalized to real-world datasets [44,45,46]. These studies mainly focus on improving detection and segmentation from high-resolution imaging. However, its applicability in enhancing PET quantification remains unexplored.

In this study, (1) we develop a new latent diffusion model for resolution recovery (LDM-RR) in PET imaging. Instead of training the diffusion model to minimize loss on the noise scale, we introduce a composite loss function with three terms: L_1_, L_2_, and MS-SSIM at the noise and image scale to improve MRI-guided reconstruction. (2) We developed a synthetic data generation pipeline to generate PET digital phantoms mimicking high-resolution PET scans for model training. (3) We evaluate the performance of our LDM-RR model in improving the statistical power of detecting longitudinal changes. (4) We evaluate the ability of the LDM-RR model to improve the agreement of amyloid PET measurements acquired using different tracers.

## 2. Materials and Methods

### 2.1. Datasets and Simulation Procedure

Imaging data from three different cohorts were used in this study to enable our experiments: (1) the Alzheimer’s Disease Neuroimaging Initiative (ADNI) cohort (adni.loni.usc.edu) [47], (2) the Open Access Series of Imaging Studies-3 (OASIS-3) [48], and (3) the Centiloid Project florbetapir (FBP) calibration dataset [49] (http://www.gaain.org/centiloid-project, accessed on 15 October 2023). A subset of the ADNI database containing MRI scans was utilized for data simulation to train the diffusion model, while another subset with FBP scans (Table 1) was employed to evaluate the model’s performance in detecting longitudinal changes. Additionally, paired FBP-PiB imaging data from the OASIS-3 and Centiloid databases (Table 1) were used to further assess the model’s performance in cross-tracer harmonization. Details regarding data selection and simulation are provided in the subsequent sections. The ADNI was launched in 2003 as a public–private partnership led by Principal Investigator Michael W. Weiner, MD. The original goal of ADNI was to test whether serial magnetic resonance imaging (MRI), positron emission tomography (PET), other biological markers, and clinical and neuropsychological assessment can be combined to measure the progression of mild cognitive impairment (MCI) and early Alzheimer’s disease (AD). The current goals include validating biomarkers for clinical trials, improving the generalizability of ADNI data by increasing diversity in the participant cohort and providing data concerning the diagnosis and progression of Alzheimer’s disease to the scientific community.

#### 2.1.1. Data to Train the Diffusion Model

We utilized 3376 MRI scans randomly selected from the ADNI database to generate simulated high-resolution digital phantoms (simDP) and simulated florbetapir (simFBP) using an MR-based procedure as previously described [50] and mimics the distribution of florbetapir (FBP) uptake in participants across a wide range of amyloid burden and clinical status and the noise and spatial resolution characteristics of typical PET images. The specific set of MRIs selected as the basis for simulation does not have a major impact on subsequent experiments and, therefore, was not described in detail. The size of the dataset captures the overall distribution and variability of structural brain differences in the elderly population without losing generalizability. A detailed description of the simulation procedure is discussed in Section 2.2 below. From this simulated dataset, 3038 samples were used to train, and 338 samples were used to validate our LDM-RR model’s performance.

#### 2.1.2. Data for Evaluating Longitudinal Power

To evaluate LDM-RR’s ability to improve statistical power to detect longitudinal changes in amyloid, we selected 167 ADNI participants with a mean age of 74.1 years (SD = 6.8), who are amyloid positive at baseline using a Centiloid cutoff of 20 [51] and have two-year follow-up (2.0 ± 0.06 years interval) FBP scans. The choice of these participants is to ensure they are on a trajectory to accumulate amyloid during the study period, i.e., having a positive expected rate of amyloid accumulation. Additional demographic information of this cohort is summarized in Table 1.

#### 2.1.3. Data for Evaluating Harmonization Performance

From the OASIS-3 database [48], we identified 113 pairs of FBP-PiB scans with a mean age of 68.1 years (SD = 8.7), and similarly, 46 pairs from the Centiloid project [49] (http://www.gaain.org/centiloid-project, accessed on 15 October 2023) with a mean age of 58.4 years (SD = 21.0). Refer to Table 1 for demographic information of these two cohorts. Studies for the cohorts included here were approved by their corresponding institutional review boards and written informed consent was obtained for each participant.

### 2.2. Image Analysis and Simulation

FreeSurfer v7.3 [52] (Martinos Center for Biomedical Imaging, Charlestown, MA, USA) (https://surfer.nmr.mgh.harvard.edu/fswiki, accessed on 15 October 2023) was used to automatically segment T1-weighted MRIs to define the anatomical regions of interests (as defined in the wmparc.mgz file). PET images were processed using a FreeSurfer-dependent pipeline that included resolution harmonization filtering, inter-frame motion correction, target frame summation, PET-to-MR registration, and regional and voxel-wise SUVR calculation [50,53]. A mean cortical SUVR (MCSUVR) was calculated as the summary measure of amyloid burden and used to evaluate longitudinal and harmonization performance [53]. For comparison purposes, a Richardson–Lucy algorithm was adopted for resolution recovery (RL-RR) through iterative deconvolution [54,55]. In our experiment, the MATLAB (The Mathworks, Inc., v2021a, Natick, MA, USA) function: deconvlucy (https://www.mathworks.com/help/images/ref/deconvlucy.html, accessed on 15 October 2023) was called with 20 iterations and an 8 mm full-width-half-max (FWHM) Gaussian kernel to generate the deconvolved high-resolution PET images and the corresponding MCSUVR estimation.

Similar to previously described by Su et al. [50], the simulation of high-resolution digital phantom (DP) and PET images (simFBP) was performed using segmented MRI as the input. For DP generation, each voxel was assigned a specific intensity value according to tissue type-specific distributions observed from actual FBP SUVR images across the aging and AD spectrum. For non-brain voxels, i.e., those not defined in the wmparc.mgz file, the voxel intensity was assigned by randomly scaling the normalized T1-MRI images to simulate moderate non-brain uptake. To generate simFBP images, the DP was smoothed and projected to the sinogram space, adding Poisson noise, and reconstructed back to the image space. We generated the simulation with a range of noise levels as seen in real-world PET scans with a noise equivalent count rate (NECR) of 75 ± 26 kcps [56,57]. The target resolution of the simFBP data is 8 mm in FWHM, approximating the resolution of standardized PET data from ADNI [4]. Figure 1 shows a visual example of a simulated digital phantom (B) and PET image (C) matching a T1-MRI image (A).

### 2.3. LDM-RR: PET Resolution Recovery Framework

We use a latent diffusion model to generate synthetic high-resolution FBP scans given standard low-resolution FBP and matching MRI scans. Figure 2 and Figure 3 give an overview of the training process. Diffusion models have shown impressive results in generating 2D images [58]. However, they are computationally demanding at the training and inference stages. Medical imaging modalities, such as MRI and PET, are more complex as they capture spatial information in 3D. Latent diffusion models operate at a lower-dimensional latent space by compressing useful information from these high-dimensional imaging data.

Our proposed LDM-RR is built upon a state-of-the-art LDM originally proposed to generate 3D brain MRIs [40]. Specifically, it has a 2-stage training process and three different components: an encoder, a diffusion U-Net [59], and a decoder model. The encoder compresses high-dimensional data into a low-dimensional latent representation, diffusion U-Net converts simFBP to simDP in the latent space through a denoising process, and the decoder upsamples the low-dimensional simDP to its original image space. Trained models and implementation code will be made available for reproducibility and further research (https://github.com/jaygshah/LDM-RR, accessed on 5 October 2024).

#### 2.3.1. Compression Models

The goal of the compression model is to create a compressed representation of high-dimensional brain images that serve as the foundation for the subsequent diffusion model. We use an autoencoder [40] that compresses the 3D brain images into a lower-dimensional latent representation capturing perceptual representation of original images while preserving essential features to reduce complexity. In the first stage, we train three modality-specific 3D autoencoder models separately for simFBP, simDP, and MRI (see Figure 2), each with a combination of L_1_ loss, perceptual loss, a patch-based adversarial objective, and a KL regularization of the latent space [40]. The input to the encoder is a 3D image with dimensions 256 × 256 × 256, and we extract smaller sub-volumes of size 64 × 64 × 64 to fit in GPU memory. The encoder maps these sub-volumes to a latent representation of size 16 × 16 × 16. Once trained, latent representations from these encoders are used as inputs to the diffusion U-Net. See Appendix B for more details on the autoencoder model architectures and hyperparameters used.

#### 2.3.2. Diffusion Model

Diffusion U-Net in LDMs perform denoising by iteratively predicting and removing noise in latent space. Typically, they are trained to minimize the L_2_ loss between predicted and actual noise [40,59]. However, for super-resolution, we found that minimizing L_2_ loss does not consistently guarantee the recovery of brain structure information in generated outputs, which we have further analyzed in the discussion section. Prior studies have shown that using a mix of image restoration loss can produce high-fidelity images compared to single loss functions [60]. L_2_ regularization is sensitive to outliers and can introduce visual artifacts since it penalizes high errors. L_1_, on the other hand, is robust to outliers but suffers from non-differentiability in zero and slow training [61]. Moreover, Zhao et al. [60] showed for image restoration and SR, L_1,_ and L_2_ penalties fail to capture structure information and proposed a multi-scale structural similarity index (MS-SSIM) metric. Voxel-level intensity has a high impact on PET quantification [62]. Here, we hypothesize and show through results that existing L_2_ loss-based diffusion models do not provide a clinically accurate reconstruction of PET scans. A weighted combination of L_1_, L_2,_ and MS-SSIM losses, on the image and noise scales can accurately generate a high-resolution using MRI and simFBP.

To train diffusion models, a small amount of Gaussian noise is progressively added to the data in T steps through a forward noise addition process, forming a Markov chain (Equations (1) and (2)) [35]:(1)$$q\left(z_{1:T}|z_{0}\right)=\prod_{t=1}^{T}q\left(z_{t}|z_{t-1}\right)$$
(2)$$q\left(z_{t}|z_{t-1}\right)=N(z_{t};\;\sqrt{1-\beta_{t}}z_{t-1},\beta_{t}I)$$

Here, $\beta_{t}$ is the fixed variance schedule and $z_{t}$ follows a pure Gaussian noise distribution after many forward diffusion steps T (T = 1000 in our experiments). The diffusion U-Net learns the reverse diffusion process, i.e., denoising $z_{T}$ to $z_{0}$ (Equations (3) and (4)) which can be formulated as [35]:(3)$$p_{\theta }\left(z_{0:T}\right)=p(z_{t})\prod_{t=1}^{T}p_{\theta }\left(z_{t-1}|z_{t}\right)$$
(4)$$p_{\theta }\left(z_{t-1}|z_{t}\right)=N(z_{t-1};\;\mu_{\theta }\left(z_{t},t\right),\;\sigma_{t}^{2}I)$$
(5)$$z_{t}=\sqrt{\alpha_{t}}z_{0}+\sqrt{1-\alpha_{t}}\epsilon$$
where $\mu_{\theta }$ represents the denoising neural network (diffusion Unet) and $\sigma_{t}^{2}=\frac{1-{\bar{\alpha }}_{t-1}}{1-{\bar{\alpha }}_{t}}\beta_{t}$. Traditionally, the diffusion models are trained to predict the added noise in forward diffusion process by minimizing L_2_ loss between predicted ($\hat{\epsilon }$) and added noise (ϵ) formulated as [59]:(6)$$L_{\theta }=E_{x,\epsilon \sim N\left(0,1\right),t}\;[{\left||\epsilon -\epsilon_{\theta }(z_{t},t)|\right|}_{2}^{2}]$$

Furthermore, we can estimate the noise-free latent vector using the predicted noise (ϵ) from the diffusion model using Equation (5) from Ho et al. [35] as:(7)$$\hat{z_{0}}=\frac{z_{t}-\sqrt{1-\alpha_{t}}\hat{\epsilon }}{\sqrt{\alpha_{t}}}$$

Zhao et al. [60] observed that image reconstruction performance can be improved by adding perceptual image metrics such as MS-SSIM in a network’s loss function. It allows capturing structural details at multiple scales while maintaining voxel-level accuracy. While this holds true for current encoder-decoder architectures, to the best of our knowledge, it has not yet been investigated for denoising diffusion networks in latent space. Since our goal is to fuse structure information from T1-MRI to guide the reconstruction, we modify LDM’s vanilla loss function ($L_{\theta }$) on noise scale to a weighted combination of L_2_ and MS-SSIM loss on image scale as:(8)$${loss}_{1}=\left(1-\alpha \right)L_{2}(z_{0},\hat{z_{0}})+\alpha MSSSIM(z_{0},\hat{z_{0}})$$

Here α=0.8 [60] in Equation (8) is an empirically set hyper-parameter. We explored α = [0.2,0.5,0.8]. However, α = 0.8 resulted in the model with the best performance in reconstructing simDP on the simulated dataset’s validation set. While L_2_ allows easier optimization in diffusion training due to its convergence properties, it is known to produce an averaging effect, which forces the model to predict values closer to the mean of training data [60]. We argue that using only L_2_ loss can help preserve whole image-level properties but may also produce inaccurate estimates at the voxel level. To this end, we propose L_1_ loss at the noise scale to ensure voxel-level details are preserved in the denoising process.
(9)$${loss}_{2}=L_{1}(\epsilon ,\hat{\epsilon })=|\epsilon -\hat{\epsilon }|$$

A combined loss function using the two loss terms from Equations (8) and (9),
(10)$${loss}_{combined}=L_{1}\left(\epsilon ,\hat{\epsilon }\right)+\left(1-\alpha \right)L_{2}(z_{0},\hat{z_{0}})+\alpha MSSSIM(z_{0},\hat{z_{0}})$$
was used to train the LDM-RR model. The combined loss is indeed equivalent to (see Appendix A):(11)$${loss}_{combined}=L_{1}\left(\epsilon ,\hat{\epsilon }\right)+{\gamma (1-\alpha )L}_{2}(\epsilon ,\hat{\epsilon })+\alpha MSSSIM(z_{0},\hat{z_{0}}))$$

By minimizing loss on image (z) and noise (ϵ) scales, the LDM-RR model learns to reduce the disparity between the reconstructed high-resolution PET image and the target digital phantom while preserving image-level and voxel-level structure details, and later (voxel-level details) may play an important role for correcting the partial volume effects.

Figure 3 illustrates the second stage of training where only the diffusion U-Net is trained whereas encoder and decoder model parameters are kept frozen. Input to the U-Net is a concatenation of noisy latent representation of simDP $\left(z_{T(DP)}\right)$ and conditioning of matching MRI $\left(z_{MR}\right)$ and simFBP $\left(z_{SP}\right)$ latent representations. The model’s predicted noise ($\hat{\epsilon }$) can be used to estimate $\hat{z_{0(DP)}}$ and calculate the combined loss (Equation (11)), which is used to update diffusion U-Net parameters in each training epoch. See Appendix B for more details on model architectures, training, and computational resources.

### 2.4. Statistical Analysis

#### 2.4.1. Simulated Data Analysis

To evaluate our LDM-RR’s actual performance at generating a high-resolution FBP scan, we compare the mean recovery coefficient (RC)—measured as the ratio of synthetic high-resolution PET MCSUVR to that of ground-truth from simulated DP, and variability measured by the standard error (SE). A value closer to 1 indicates higher reconstruction performance. We compare our method to Richardson–Lucy-based resolution recovery (RL-RR), traditional LDM with L_2_ noise loss, LDM with L_2_ noise and L_1_ image loss, and simulated FBP scans without any correction.

#### 2.4.2. Longitudinal Analysis

To evaluate the longitudinal performance of the LDM-RR, for each of the 167 participants, the annualized rate of amyloid accumulation was calculated by dividing the MCSUVR change from baseline to the follow-up visit by the follow-up interval, commonly used in longitudinal PET studies [63]. The imaging data was acquired using standard protocols, and harmonization procedures were performed to reduce variability. The mean and standard deviation of the annualized rate of change were evaluated for each analysis method, i.e., raw measurement, with RL-RR, and with the LDM-RR. A one-sample *t*-test (one tail) was used to determine whether the annualized rate was significantly greater than zero. A smaller *p*-value is interpreted as having greater power to detect the longitudinal accumulation of amyloid burden. To further compare the statistical power of different techniques in a longitudinal setting, we estimated the number of participants per arm needed to detect a 25% reduction in amyloid accumulation rate due to treatment with 80% power and a two-tailed type-I error of *p* = 0.05 in hypothetical anti-amyloid treatment trials similar to previous studies [64,65]. A smaller estimated sample size (SS) indicates greater statistical power.

#### 2.4.3. Cross-Tracer Analysis

In the cross-tracer analysis, we evaluated the impact of RL-RR vs. LDM-RR on the agreement of PET-derived global amyloid burden, i.e., MCSUVR, using paired FBP-PIB data from OASIS-3 and the Centiloid project. The imaging data was also collected following standard protocols and underwent harmonization processes to minimize variability, similar to the ADNI study. In our experiment, the raw PIB MCSUVR was used as the reference amyloid burden measurement, and we evaluated whether the corrected FBP MCSUVR is more strongly correlated with PIB MCSUVR using Steiger’s test. We also test whether the LDM-RR corrected FBP MCSUVR is more strongly associated with PIB MCSUVR than the RL method.

## 3. Results

### 3.1. Qualitative Assessments

Figure 4 showcases corrected FBP scans using RL (Figure 4C) and LDM-RR (Figure 4D) methods in comparison to the real FBP scan without any correction (Figure 4A). The proposed LDM-RR model-generated synthetic FBP image has an improved spatial resolution, with a similar level of anatomical details matching T1-MRI (Figure 4B). A similar example of LDM-RR applied to simulated data is shown in Figure 1D. Although RL-RR does not require an MRI, it generated noisier images and was not able to fully recover the high-resolution details (Figure 4C). The LDM-RR method leverages the high-resolution structural information from MRI to guide the super-resolution process, resulting in PET images with reduced partial volume effects.

### 3.2. Evaluation on Simulated Data

A visual example of model-generated synthetic FBP from the test set of simulated data is shown in Figure 1D. The mean RC from different diffusion models compared to the RL-RR method and without any correction is shown in Figure 5. Our proposed LDM-RR model was able to better reconstruct target simDP (0.96, SE = 0.004, *p* < 0.001) compared to RL-based correction (0.82, SE = 0.005, *p* < 0.001) and without any corrections (0.76, SE = 0.008, *p* < 0.001). It also performs significantly better compared to a typical LDM architecture [40] for super-resolution (1.32, SE = 0.08, *p* < 0.001) and other combinations of noise and image scale loss (1.58, SE = 0.09, *p* < 0.001). The improvement in recovery coefficient with LDM-RR was also statistically significant (*p* < 0.001) compared to other LDM methods.

### 3.3. Evaluation on Real Longitudinal Amyloid PET Data

Table 2 shows a comparison of statistical power to detect amyloid accumulation in longitudinal studies using the LDM and RL methods for resolution recovery in comparison to measurements from raw FBPs without any correction. The annualized rate of amyloid accumulation was significantly greater than zero for all three methods (*p* < 0.0001), suggesting an increase in brain amyloid burden over time as expected. Notice the annualized rate of amyloid accumulation is the unit of SUVR/year, which is specific to the underlying quantification methods and not directly comparable. Numerically, the *p*-value was smallest using the LDM-RR and largest without any correction, suggesting our proposed method had the best power in detecting longitudinal changes. Additionally, the LDM-RR required a much smaller sample size estimate to detect a 25% reduction in the amyloid accumulation rate due to treatment in hypothetical anti-amyloid trials. To put the sample size estimation into context, the recently completed TRAILBLAZER-ALZ2 randomized trial of donanemab [66] recruited 860 participants for the treatment arm and 876 for the placebo arm. The donanemab was able to reduce the patient’s brain amyloid burden by over 80%. Therefore, our assumed treatment effect is considerably more moderate.

### 3.4. Evaluation on Real Cross-Tracer Amyloid PET Data

The performance of LDM-RR and RL-RR methods at harmonizing cross-tracer global amyloid burden measurements is shown in Table 3. Agreement of MCSUVR measurements between tracers significantly improved (*p* < 0.001), as shown by a higher correlation for both LDM and RL-based corrections to the reference measure. The improvements in LDM-RR-based partial volume corrections compared to RL were also statistically significant (*p* = 0.042). The results provide evidence supporting that the inter-tracer variability in PET-derived amyloid burden measurement is at least in part related to the partial volume effect associated with lower spatial resolution and the contaminated signal from the target regions of interest. While the numerical improvement in terms of the Pearson correlation is small, they were statistically significant, suggesting improving image resolution can be one of the strategies for reducing the variability.

## 4. Discussion

It is well recognized that PET imaging has inherently low spatial resolution which leads to PVE, resulting in loss of sensitivity to focal changes and compromised accuracy due to signal contamination [14,67]. Many different techniques have been developed to account for PVE and improve quantitative accuracy [14,16,68,69]. In the context of PET neuroimaging, commonly adopted techniques are often region-based and do not provide high-resolution images [50,70,71]. Voxel-wise approaches do exist [15,16,72]; however, they are known to amplify noise while having limited ability to recover the full spatial resolution or have gone through limited evaluation for targeted applications [73,74,75]. This study presents a new approach to improving PET quantification leveraging latent diffusion models trained using controlled simulated data. We show that diffusion models have a strong potential to enhance PET quantification through super-resolution. Our LDM-RR model’s performance on longitudinal amyloid and cross-tracer PET data demonstrates that diffusion-based super-resolution (SR) approaches can outperform traditional approaches in tackling the issue of PVE in PET imaging.

We propose an alternative to L_2_ loss, which has been a de facto standard in training diffusion models. L_2_ penalty pushes the model to reduce large errors, potentially sacrificing high-frequency details (Figure 6B). Moreover, L_2_ loss is sensitive to the scale of voxel intensities. In super-resolution tasks, where the goal is to reconstruct fine details (voxel-level), L_2_ loss may not be ideal for capturing subtle differences in high-frequency information, whereas L_1_ loss may help capture the voxel-level details, which are considered to be crucial to addressing PVE. In addition, it is interesting to observe the added contribution from the multi-scale structural similarity index (MS-SSIM) metric, which confirmed the research findings of Zhao et al. [60]. Visual comparison in Figure 6 shows that using a combined loss at image and noise scales (Figure 6D), the generated high-resolution FBP images have a more accurate representation of brain structure from MRI and voxel-level uptake measurements.

Simulation data was generated to approximate the PET imaging formation process and the distribution of tracer uptake as observed in real amyloid PET images. The simulated data were used to train the LDM-RR model and evaluate its performance against the ground truth which is otherwise not possible. It should be recognized that simulated data cannot fully replicate the overall distribution and characteristics of real amyloid PET data which may introduce bias to the trained model. More sophisticated simulations can potentially be adopted to minimize this potential bias and improve model performance. Nevertheless, evaluation of model performance in real-world setting is important which we discuss further below.

We selected two commonly encountered scenarios in the investigation of Alzheimer’s disease to evaluate the real-world utility and benefits of diffusion-based SR techniques for PET. In the longitudinal analysis, we leveraged data from the ADNI cohort in participants with a baseline visit and a 2-year follow-up to examine the sensitivity and statistical power of different correction methods. The participants were intentionally selected to have moderate to medium-high levels of pathological amyloid burden at baseline to maximize the probability of these participants to accumulate amyloid plaques during the follow-up period, and therefore we expect a positive increase in the MCSUVR measure and deviation from that reflects measurement noise. It is worth noting that both resolution recovery methods (RL-RR and LDM-RR) led to greater numerical values of the rate of amyloid accumulation which reflects the improved recovery coefficient as expected. In the meantime, the standard deviation of the estimated rate also increased numerically which can be a combined effect of the improved recovery coefficient and possible amplification of noise. The net effect of the correction methods is reflected by the *p*-values of the one-sample *t*-test applied to the rate data where a smaller *p*-value indicated a greater statistical power demonstrating a beneficial effect of correcting for PVE. The sample size estimation in hypothetical anti-amyloid treatment trials further confirmed the notion that correcting for PVE improves the longitudinal power. This improvement can lead to reduced experimental costs in longitudinal observational studies and clinical trials which will facilitate treatment development. In a clinical setting this improved power can lead to better patient management by providing more sensitive and accurate monitoring of disease progression once treatment becomes routinely available to patients. While we chose to demonstrate the capability of our proposed technique to improve the quantitative analysis of clinical amyloid PET imaging data, the same principle and method can also be applied to the analysis of preclinical animal PET data. Previous studies have developed advanced algorithms for the analysis of preclinical PET data, e.g., [76]. Super-resolution methods in general and our proposed LDM-RR technique specifically can recover the high-frequency signals lost during the image formation process by leveraging other sources of information such as MR or prior knowledge such as a template and improve the quantitative accuracy of PET-derived measurements in both preclinical and clinical applications.

The second real-world application we tested in this study is the ability of PVE correction to improve agreement between PET-derived measurements from different tracers. Using amyloid PET imaging as an example, currently, there are at least five different PET tracers that are widely used in research studies, clinical trials, and patient management to measure amyloid burden. It is well recognized that the different tracers behave differently, leading to discrepancies in PET-derived amyloid burden measurements. At least part of this discrepancy is related to the contamination of the target measurement from nuance signals spill-in to our measurements. We demonstrated that both correction methods improved the agreement, and the LDM-RR outperformed the RL method statistically. In the meantime, we recognize the improvement has not reached the level where two tracers can be used interchangeably and may not outperform some of the other techniques we have developed [6,7]. On the other hand, this experiment demonstrated that PVE correction can be one of the strategies we can employ to improve harmonization, and a combination of multiple techniques may be the ultimate solution to fully solve the harmonization problem. We also acknowledge that there are other approaches leveraging deep learning techniques to address issues related to the standardization and harmonization of image-derived measurements, e.g., [77]. Further investigation in this direction is warranted.

This work has a few limitations: One potential limitation of our diffusion model-based framework is the computational complexity to train and validate the model on 3D imaging data. Even with a faster sampling method [78], the inference time is considerably high compared to other generative models (see Appendix B). Due to the sequential nature of the denoising process, this is a known limitation of diffusion models and remains an active area of research. Second, the model was trained on synthetic data rather than real data, which may limit the technique’s performance. This is primarily due to the fact that the approach we adopted requires paired data with ground truth high-resolution images for training and validation, which is lacking. Examination of methods that are self-supervised or semi-supervised may allow us to overcome this limitation. We also like to point out that our intended application is narrowly focused on AD-related applications, while the underlying principles can be applied more broadly, although it will be beyond the scope of this paper.

## 5. Conclusions

We introduced a latent-diffusion-model-based resolution recovery (LDM-RR) method to enhance PET image resolution and mitigate the impact of PVE. Results demonstrate that the LDM-RR method improves spatial resolution while preserving critical amyloid and anatomical information, outperforming traditional methods like Richardson–Lucy (RL) correction. LDM-RR model showed superior performance at reconstructing high-resolution PET images, improved statistical power for detecting longitudinal amyloid accumulation and a strong potential to improve the agreement between measurements obtained from different PET tracers, contributing to better data harmonization across multi-center studies. These findings suggest that diffusion-based super-resolution (SR) techniques offer a promising alternative to conventional PVC methods by overcoming noise amplification issues and achieving better image fidelity.

## Figures and Tables

**Figure 1 life-14-01580-f001:**
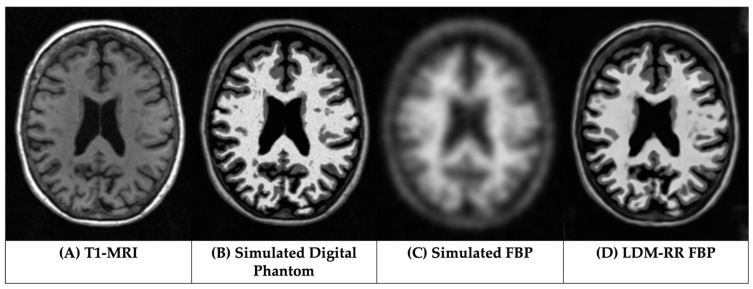
Visualization of a simulated digital phantom (simDP) and simulated FBP (simFBP) from the data simulation pipeline using T1-MRI and the LDM-RR generated synthetic super-resolution FBP.

**Figure 2 life-14-01580-f002:**
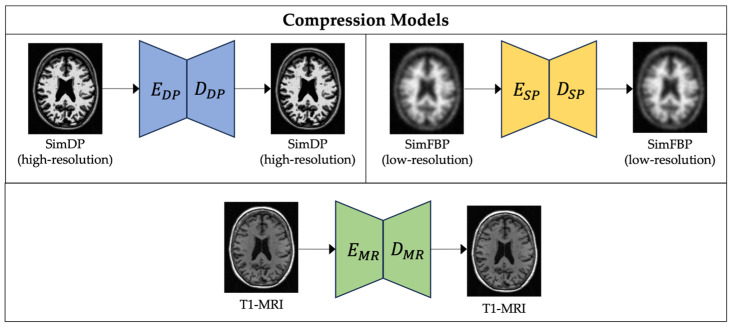
Training three modality-specific autoencoder models to compress high-dimensional simulated DP, simulated FBP and MRI data into a lower-dimensional latent representation.

**Figure 3 life-14-01580-f003:**
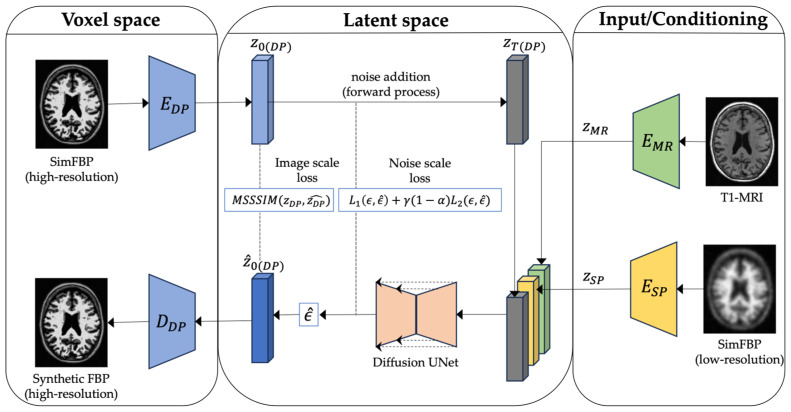
Proposed LDM-RR framework’s training process for PET super-resolution. LDM is conditioned on latent representations of T1-MRI and simFBP and uses a combination of image and noise scale losses to generate corresponding high-resolution simDP.

**Figure 4 life-14-01580-f004:**
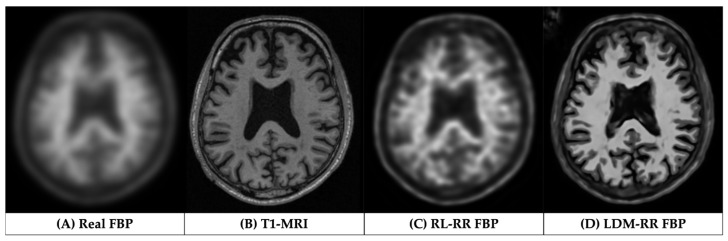
Visual comparison of generated FBP scans using RL-RR and our LDM-RR to real FBP and T1- MRI for a sample from OASIS-3 cohort.

**Figure 5 life-14-01580-f005:**
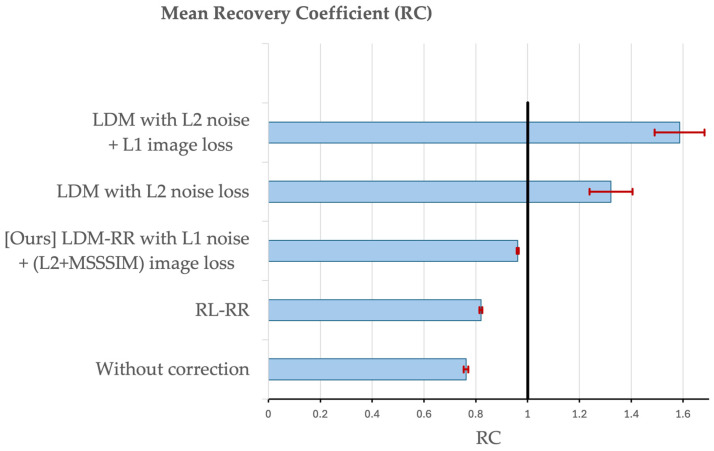
Comparison of mean recovery coefficient (RC) using different methods on a held-out test of 338 samples randomly selected from the simulated dataset. A value closer to 1 indicates high performance.

**Figure 6 life-14-01580-f006:**
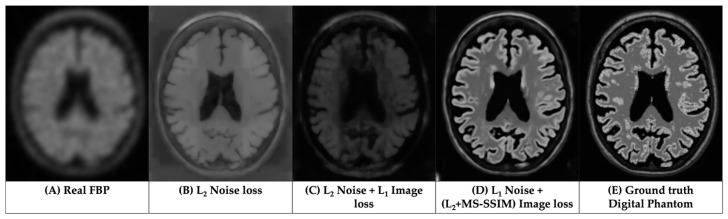
Comparisons of the results from LDM trained using different loss functions.

**Table 1 life-14-01580-t001:** Summary of demographic information of the three cohorts included in this study. * 14 out of 46 were unknown.

Cohort	ADNI	OASIS-3	Centiloid
**Sample count**	334 (167 baseline-followup FBPs)	113 (FBP-PIB pairs)	46 (FBP-PIB pairs)
**Age (SD)** **years**	75.1 (6.9)	68.1 (8.7)	58.4 (21.0)
**Education (SD)** **years**	16.1 (2.7)	15.8 (2.6)	NA
**Male (%)**	182 (54.5%)	48 (42.5%)	27 (58.7%)
**Cognitive** **impairment (%)**	236 (70.6%)	5 (4.4%)	24 (52.2%)
**APOE4+ (%)**	218 (65.3%)	38 (33.6%)	15 (46.9 *%)
**PET interval (SD)** **years**	2.0 (0.06)	NA	NA

**Table 2 life-14-01580-t002:** Statistical power in detecting longitudinal changes measured by mean, standard deviation, and *p*-value of an annualized rate of amyloid accumulation and sample size (SS) per arm estimates detecting a 25% reduction in amyloid accumulation rate due to treatment (80% power and a two-tailed type-I error of *p* = 0.05).

Annualized Rate	Raw	RL-RR	LDM-RR
Mean	0.0278	0.0377	0.0459
SD	0.0664	0.0807	0.0881
*p*-value	1.0 × 10^−7^	5.0 × 10^−9^	1.3 × 10^−10^
SS	1431	1154	926

**Table 3 life-14-01580-t003:** Comparison of RL and LDM-RR methods in improving the MCSUVR agreement between FBP and PIB tracers shown by Pearson correlation and Steiger’s test.

Method	Pearson Correlation	Steiger’s *p*-Value
**Without correction**	0.9163	N/A
**RL-RR**	0.9308	<0.0001 (RL-RR vs. without correction)
**LDM-RR**	0.9411	0.0001 (LDM-RR vs. without correction)
		0.0421 (LDM-RR vs. RL-RR)

## Data Availability

This study utilized datasets obtained from existing sources including ADNI, OASIS-3, and the Centiloid Project. ADNI data used in this study is available upon registration and compliance with data use agreements through the ADNI database (www.adni.loni.usc.edu). OASIS-3 data was obtained from the OASIS-3 database, accessible upon registration and approval through the OASIS project website (www.oasis-brains.org). The Centiloid data is available for qualified researchers upon request through the Centiloid Project (www.gaain.org/centiloid-project).

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
