# Peer review of "Enhancing Amyloid PET Quantification: MRI-Guided Super-Resolution Using Latent Diffusion Models"

_life, 2024, doi:10.3390/life14121580_

Round 1
Reviewer 1 Report
Comments and Suggestions for Authors
In this manuscript (life-3286540), the Authors introduced a latent diffusion model-based resolution recovery method to improve PET image resolution and mitigate the impact of partial volume effects. The present work is significant for the diagnosis of Alzheimer's, particularly the non-invasive detection of amyloid-β plaques in the brain via amyloid PET imaging. Overall, this study is cutting-edge and well-organized. Given the evidence provided, I believe the conclusions are sensible and do not over-reach. I suggest the study be published in Life Journal after addressing the following minor issues.
1. In the subsection 2.1. Datasets and Simulation Procedure, besides ADNI, the rationale for using imaging data cohorts from the Open Access Series of Imaging Studies-3 (OASIS-3) and the Centiloid Project florbetapir calibration dataset should also be explained.
2. If possible, the protocol number of IRB/ethical clearance regarding studies for the cohorts could be provided in the Materials and Methods Section.
3. The discussion of Figure 4A is somehow missing.
4. As described in [50] (line 178). The authors of ref. [50] should be mentioned properly in the sentence.
5. All equations should be numbered accordingly and mentioned in the text.
6. The use of uppercase and lowercase letters throughout the manuscript should be consistent.
7. The manuscript has a few grammatical errors that should be revisited and corrected such as measurememts (line 382) etc.
8. All references should be in a consistent style and format.
Author Response
We thank the reviewer for their insightful comments. We have revised our manuscript and provided a point-by-point response to the reviews below. In addition, we have made all changes to the manuscript using the “track changes” function.
- In the subsection 2.1. Datasets and Simulation Procedure, besides ADNI, the rationale for using imaging data cohorts from the Open Access Series of Imaging Studies-3 (OASIS-3) and the Centiloid Project florbetapir calibration dataset should also be explained.
Response: We thank the reviewer for this feedback. We have now clarified this and added the following sentences in subsection 2.1 [Lines 128-134]:
“A subset of the ADNI database containing MRI scans was utilized for data simulation to train the diffusion model, while another subset with FBP scans (Table 1) was employed to evaluate the model's performance in detecting longitudinal changes. Additionally, paired FBP-PiB imaging data from the OASIS-3 and Centiloid databases (table 1) were used to further assess the model's performance in cross-tracer harmonization. Details regarding data selection and simulation are provided in the subsequent sections.”
- If possible, the protocol number of IRB/ethical clearance regarding studies for the cohorts could be provided in the Materials and Methods Section.
Response: We thank the reviewer for this feedback. In this study, we only performed secondary analyses of data obtained from several different sources. Unfortunately, the IRB protocol number is not commonly published, and we do not know the number. - The discussion of Figure 4A is somehow missing.
Response: We thank the reviewer for pointing this out. We have now updated lines 363-364 in the manuscript to reflect the description of Figure 4A as follows:
“Figure 4 showcases corrected FBP scans using RL (Figure 4C) and LDM-RR (Figure 4D) methods in comparison to the real FBP scan without any correction (Figure 4A)” - As described in [50] (line 178). The authors of ref. [50] should be mentioned properly in the sentence.
Response: We thank the reviewer for pointing this out. We have made edits to the manuscript accordingly (now line 190). - All equations should be numbered accordingly and mentioned in the text.
Response: Yes, we agree. We have labeled all equations in the manuscript and updated the text accordingly. - The use of uppercase and lowercase letters throughout the manuscript should be consistent.
Response: Thanks for pointing this out. We have reviewed the manuscript for consistency in uppercase/lowercase letters and made corrections as needed. - The manuscript has a few grammatical errors that should be revisited and corrected such as measurememts (line 382) etc.
Response: We thank the reviewer for pointing this out. We have rectified the typo in line 382 and other grammatical errors throughout the manuscript with track changes enabled for quick review. - All references should be in a consistent style and format.
Response: We thank the reviewer for this suggestion. We have updated references such that they are consistent with the style and format of the journal template.
Reviewer 2 Report
Comments and Suggestions for Authors
The title “Enhancing PET Quantification: MRI-guided Super-Resolution using Latent Diffusion Models” for the manuscript should be revised to include information regarding the target disease to become more appropriate for the manuscript.
The manuscript discusses the development of a novel latent diffusion model for resolution recovery (LDM-RR) in PET imaging. It is based on the introduction of a composite loss function with three terms: L1, L2, and multi-scale structural similarity index (MS-SSIM) on the noise and image scales to improve MRI-guided reconstruction. A synthetic data generation pipeline was developed to generate digital PET models that mimic high-resolution PET scans for model training. Performance evaluation of the model was conducted.
Overall the article is acceptable, but it should be improved, following the comments below as it slightly lacks translationality.
The comments provide further details.
The authors are requested to improve the state of the art, in particular to improve the description of the advantages and limitations of this method, in the discussion section and in the conclusions.
Authors are requested to create an applicative aspect of this method also in the preclinical field. Authors are suggested to consider [10.1007/978-3-031-13321-3_31] in which the authors apply a semi-automatic segmentation process in the preclinical field based on the PET image atlas performed in the native space and in the model space associated with radiomic repeatability in mouse models. Authors are strongly advised to carve out within the manuscript an applicability, if potentially foreseen, to preclinical PET imaging since all oncology and neurological experimentation always passes through a workflow that goes from in vitro tests to in vivo tests, up to the clinic both in diagnosis and in therapy. In this suggested work the authors develop a radiomics workflow based on PET/CT Image Acquisition and Segmentation, Warping Atlas to CT, Warping CT to Atlas, Extraction of Radiomics Features. It is strongly recommended that the authors introduce statistical performance parameters following the one applied in this work and compare the applicability of their method to in vivo studies. Rinconsidering this kind of scientific experiment, this would make the manuscript more versatile and innovative.
Authors may consider [10.1007/s11831-024-10067-w] where the possibility of accurate image standardization and harmonization models to mitigate image variability obstacles is discussed extensively. The manuscript is based on three-dimensional convolutional neural networks applied to kidney images. The integration of deep learning frameworks and volumetric segmentation algorithms can effectively improve disease diagnosis accuracy, especially cysts, and monitoring clinical development in individuals with various clinical conditions.
Figures 5 and 6 appear low resolution. Please replace them with high resolution figures. Also please respect the original template format for figure captions.
To make reading the document easier for the reader, an Abbreviations section should be added to the end of the document.
In conclusion, to make the article more coherent, the authors should expand the references and improve the English to make the discourse more fluent throughout the document.
Comments on the Quality of English LanguageAthors should improve the English to make the discourse more fluent throughout the document.
Author Response
We thank the reviewer for their insightful comments. We have revised our manuscript and provided a point-by-point response to the reviews below. In addition, we have made all changes to the manuscript using the “track changes” function.
- The title “Enhancing PET Quantification: MRI-guided Super-Resolution using Latent Diffusion Models” for the manuscript should be revised to include information regarding the target disease to become more appropriate for the manuscript.
Response: We thank the reviewer for this suggestion. We have updated the manuscript title to “Enhancing Amyloid PET Quantification: MRI-guided Super-Resolution using Latent Diffusion Models” which more accurately captures the application of the proposed framework.
- The authors are requested to improve the state of the art, in particular to improve the description of the advantages and limitations of this method, in the discussion section and in the conclusions.
Response: We made major revisions to the first paragraph of the discussion to provide additional context and highlight the state of the art, incorporating five more relevant prior studies [Lines 426-433]. The limitation of our specific method is also discussed in the last paragraph of the discussion [Lines 518-525]. The conclusion section is slightly updated while we focus on the specific methods and applications investigated in this paper.
- Authors are requested to create an applicative aspect of this method also in the preclinical field.
Response: We thank the reviewer for the suggestion. The following sentences were added to the discussion section [Lines 486-494]:
“While we chose to demonstrate the capability of our proposed technique to improve quantitative analysis of clinical amyloid PET imaging data, the same principle and method can also be applied to the analysis of preclinical animal PET data. Previous studies have developed advanced algorithms for the analysis of preclinical PET data, e.g., [76]. Super-resolution methods, in general, and our proposed LDM-RR technique specifically can recover the high-frequency signals lost during the image formation process by leveraging other sources of information such as MR or prior knowledge such as a template and improve the quantitative accuracy of PET-derived measurements in both preclinical and clinical applications.”
We hope the reviewer finds the addition sufficient.
- Authors are suggested to consider [10.1007/978-3-031-13321-3_31] in which the authors apply a semi-automatic segmentation process in the preclinical field based on the PET image atlas performed in the native space and in the model space associated with radiomic repeatability in mouse models. Authors are strongly advised to carve out within the manuscript an applicability, if potentially foreseen, to preclinical PET imaging since all oncology and neurological experimentation always passes through a workflow that goes from in vitro tests to in vivo tests, up to the clinic both in diagnosis and in therapy. In this suggested work the authors develop a radiomics workflow based on PET/CT Image Acquisition and Segmentation, Warping Atlas to CT, Warping CT to Atlas, Extraction of Radiomics Features. It is strongly recommended that the authors introduce statistical performance parameters following the one applied in this work and compare the applicability of their method to in vivo studies. Rinconsidering this kind of scientific experiment, this would make the manuscript more versatile and innovative.
Response: We thank the reviewer for this suggestion. We have cited the suggested work. The changes were reflected in our response to the previous comment (number 3).
Regarding the statistical tests, we performed and reported statistical test results specific to each of our experiments. We have now updated section 3.2 with results on simulated data with paired t-test results [Lines 378-384]. We hope this addresses the reviewer’s concern.
- Authors may consider [10.1007/s11831-024-10067-w] where the possibility of accurate image standardization and harmonization models to mitigate image variability obstacles is discussed extensively. The manuscript is based on three-dimensional convolutional neural networks applied to kidney images. The integration of deep learning frameworks and volumetric segmentation algorithms can effectively improve disease diagnosis accuracy, especially cysts, and monitoring clinical development in individuals with various clinical conditions.
Response: We thank the reviewer for this suggestion. We have cited this work and added two sentences in the discussion section highlighting its relevance to our current work [Lines 509-512].
“We also acknowledge that there are other approaches leveraging deep learning techniques to address issues related to the standardization and harmonization of image-derived measurements, e.g. [77]. Further investigation in this direction is warranted.”
- Figures 5 and 6 appear low resolution. Please replace them with high resolution figures. Also please respect the original template format for figure captions.
Response: We thank the reviewer for pointing this out. We have provided high-resolution figure images (330 dpi) in a zip folder to the journal, and they will be used in the production stage. Also, we have updated the format of the figure and table captions in the manuscript to match the original template.
- To make reading the document easier for the reader, an Abbreviations section should be added to the end of the document.
Response: We thank the reviewer for this suggestion. We have now added a Table with a list of abbreviations and their definitions at the end of the document as Appendix C.
- In conclusion, to make the article more coherent, the authors should expand the references and improve the English to make the discourse more fluent throughout the document.
Response: We thank the reviewer for this feedback. We have expanded the references and improved the writing throughout the manuscript as recommended.
Reviewer 3 Report
Comments and Suggestions for Authors
General comment:
This work deals with the PET imaging of amyloid plaques, supported by morphological information from MRI, in the brain for AD diagnosis. The data are synthetic and simulated on phantoms, plus the aid of machine learning methods.
The approach is interesting, but the results may be considered to be preliminary and the manuscript demands revisions.
Specific comments throughout the paper:
The introduction is pretty focused and discuss clearly the problem and knowledge gap of proposing a partial volume correction method alternative to tose found in the literature.
The dataset is poorly commented and the potential shortcomings (e.g., homogeneity and uniformity of the data and images, as well as differences in the imaging apparatuses) are not described and discussed. This is rather crucial for the aim and scope of the authors.
The procedure for using the >3k MRI scans to generate phantoms is not clear (lines 138-142).
Line 142: what is the value the authors used as noise of PET images? This is not reported, please be quantitative.
The dataset variabilities is probably too much and how it has been handled by the authors for evaluating the longitudinal power and harmonization performance is not clear enough. Please revise these two subsections.
Line 175: please provide reference (or links) to the deconvlucy function.
Thanks for providing a link for the code, this is remarkable (line 208).
The compression model is not explained clearly, more details are needed. In my opinion the readership can get lost easily.
Equations are not numbered.
Moreover, some symbols are not explained properly (e.g., what the sub-script means). Please revise. This is not understandable nor reproducible.
Missing ref for Eq. at line 281.
Line 283: why not optimizing the hyperparameter? A discussion about this point is in order. This is very relevant for the validity and robustness of your method and results.
Fig. 5 is not of good quality. Please improve it.
The results section is not of enough quality. It is very short, with poor comments and not so many findings. This is a shortcoming.
The discussion section is appreciated, but there isn’t a fair comparison with the literature and reference to other methodologies and ground truths. Therefore, per se, the study is limited.
Minor issues:
Line 149: missing space
Lines 164-177: not aligned as per the template
Please do not indent after equations, follow the template guidelines.
Author Response
We thank the reviewer for their insightful comments. We have revised our manuscript and provided a point-by-point response to the reviews below. In addition, we have made all changes to the manuscript using the “track changes” function.
- The introduction is pretty focused and discuss clearly the problem and knowledge gap of proposing a partial volume correction method alternative to those found in the literature.
Response: We thank the reviewer for this positive feedback and compliments for our work. - The dataset is poorly commented and the potential shortcomings (e.g., homogeneity and uniformity of the data and images, as well as differences in the imaging apparatuses) are not described and discussed. This is rather crucial for the aim and scope of the authors.
Response: We thank the reviewer for this comment. The imaging datasets we used were from studies that adopted standard acquisition protocols, and image harmonization has been performed following established protocols [Weiner, Michael W., et al. "The Alzheimer's Disease Neuroimaging Initiative 3: Continued innovation for clinical trial improvement." Alzheimer's & Dementia (2017)]. We have now clarified this in the manuscript in section 2.4.2 [Lines 337-339] and in section 2.4.3 [Lines 354-356].
- The procedure for using the >3k MRI scans to generate phantoms is not clear (lines 138-142).
Response: We thank the reviewer for this comment. Although section 2.1.1. focus on describing the data we used for data simulation; we have now explained the simulation procedure based on Su et al. [50] in more detail in section 2.2. We have added the following sentences in the manuscript [Lines 150-154]:
“The specific set of MRIs selected as the basis for simulation does not have a major impact on subsequent experiments and, therefore, was not described in detail. The size of the dataset captures the overall distribution and variability of structural brain differences in the elderly population without losing generalizability. A detailed description of the simulation procedure is discussed in section 2.2 below.”
- Line 142: what is the value the authors used as noise of PET images? This is not reported, please be quantitative.
Response: We thank the reviewer for this feedback. We have updated the manuscript to incorporate this information in section 2.2 with appropriate references to existing studies. We added the following sentence [Lines 198-199]:
“We generated the simulation with a range of noise level as seen in real world PET scans with noise equivalent count rate (NECR) of 75±26 kcps [56,57].”
- The dataset variabilities is probably too much and how it has been handled by the authors for evaluating the longitudinal power and harmonization performance is not clear enough. Please revise these two subsections.
Response: We thank the reviewer for this feedback. To address the concerns related to variabilities in imaging datasets for longitudinal and cross-tracer analysis, we have now updated the section 2.4.2 [Lines 338-339] and 2.4.3 [Lines 354-356] to include the following sentence for clarification:
“The imaging data was collected following standard protocols and underwent harmonization processes to minimize variability.”
Further, we added a citation to an existing study [Bollack, Ariane, et al., "Investigating reliable amyloid accumulation in Centiloids: Results from the AMYPAD Prognostic and Natural History Study." Alzheimer's & Dementia (2024)], which provides a reference to the metric we have used for longitudinal analysis.
- Line 175: please provide reference (or links) to the deconvlucy function.
Response: We thank the reviewer for this suggestion. We have updated the manuscript with a link to MATLAB’s deconvlucy function for reference.
- Thanks for providing a link for the code, this is remarkable (line 208).
Response: We thank the reviewer for this compliment.
- The compression model is not explained clearly, more details are needed. In my opinion the readership can get lost easily.
Response: We thank the reviewer for this comment. We have updated section 2.3.1, which now better describes the role of compression models in our framework. Further details of its architecture and hyperparameters are included in Appendix B. We have added the following sentences [Lines 226-230]:
“The goal of the compression model is to create a compressed representation of high-dimensional brain images that serve as the foundation for the subsequent diffusion model. We use an autoencoder [40] that compresses the 3D brain images into a lower-dimensional latent representation capturing perceptual representation of original images while preserving essential features to reduce complexity.”
- Equations are not numbered.
Response: We thank the reviewer for pointing this out. We have labeled all the equations in the manuscript and have updated references to those equations.
- Moreover, some symbols are not explained properly (e.g., what the sub-script means). Please revise. This is not understandable nor reproducible.
Response: We thank the reviewer for pointing this out. We have now updated the manuscript with Appendix C, which has a list of abbreviations used in the manuscript along with its definition for better readability.
- Missing ref for Eq. at line 281.
Response: We thank the reviewer for this feedback. The equation in line 281 (now equation 8) is part of the proposed loss function in this study. This is based on a modification of the original latent diffusion model’s loss objective from equation 6, as we have described in the manuscript. We have provided a reference to a study by Ho et al. [35] that describes in further detail. We hope this addresses the reviewer’s concern.
- Line 283: why not optimizing the hyperparameter? A discussion about this point is in order. This is very relevant for the validity and robustness of your method and results.
Response: We thank the reviewer for this feedback. Indeed, we explored hyperparameter tuning with =0.2, 0.5, and 0.8. However, =0.8 resulted in the model with the best performance on the simulated dataset’s validation set, which was consistent with the findings of an earlier study by Zhao et al. [57]. We have added the following sentence for clarity [Lines 292-294]:
“We explored α=[0.2,0.5,0.8]. However, α=0.8 resulted in the model with the best performance in reconstructing simDP on the simulated dataset’s validation set.”
- Fig. 5 is not of good quality. Please improve it.
Response: We thank the reviewer for pointing this out. We have now provided high-resolution figure images (330 dpi) in a zip folder to the journal, and they will be used in the production stage.
- The results section is not of enough quality. It is very short, with poor comments and not so many findings. This is a shortcoming.
Response: We thank the reviewer for this feedback. We have now substantially expanded our results section and included more detailed descriptions and explanations to the results section.
- The discussion section is appreciated, but there isn’t a fair comparison with the literature and reference to other methodologies and ground truths. Therefore, per se, the study is limited.
Response: We thank the reviewer for this feedback. We have now updated the discussion section and compared our approach to two existing segmentation-based studies to improve imaging analysis [refs 76, 77]. Kindly see Lines 486-494 and Lines 509-512. Additionally, we have also now expanded on the limitations of our proposed approach. We hope this addresses the reviewer’s concern [Lines 518-525].
- Minor issues:
- Line 149: missing space
- Lines 164-177: not aligned as per the template
- Please do not indent after equations, follow the template guidelines.
Response: We thank the reviewers for these comments. We have now updated the manuscript to reflect these changes.
Round 2
Reviewer 2 Report
Comments and Suggestions for Authors
The authors responded comprehensively to all comments. In addition, the authors have extensively refined the manuscript, and its impact may now reach a wider audience.
Author Response
Comments1: The authors responded comprehensively to all comments. In addition, the authors have extensively refined the manuscript, and its impact may now reach a wider audience.
Response1: We thank the reviewer for this kind and encouraging feedback.
Reviewer 3 Report
Comments and Suggestions for Authors
I sincerely thanks the authors for thoroughly revising their work and replying to my questions and doubts, that have been take into account in the modification process.
Some minor issues remains. Please fix them.
Comments on the Quality of English LanguageThe english can be improved.
Author Response
Comments1: I sincerely thanks the authors for thoroughly revising their work and replying to my questions and doubts, that have been take into account in the modification process.
Some minor issues remains. Please fix them.
Response1: We thank the reviewer for this feedback. We have updated the manuscript by removing punctuation errors and inconsistencies with word capitalization in section headings using the track changes feature in Word. We hope this addresses the reviewer's concern.